



**Integrated soil fertility management drives the effect of cover crops on GHG**
**emissions in an irrigated field**
*Guillermo Guardia[a]\*, Diego Abalos[b], Sonia García-Marco[a], Miguel Quemada[a], María Alonso-*
*Ayuso[a], Laura M. Cárdenas [c], Elizabeth R. Dixon[c], Antonio Vallejo[a]*
[a] ETSI Agronomos, Technical University of Madrid, Ciudad Universitaria, 28040 Madrid, Spain.
[b] School of Environmental Sciences, University of Guelph, Guelph, Ontario, N1G 2W1, Canada.
[c] Rothamsted Research, North Wyke, Devon, EX20 2SB, UK.
* Corresponding author. Tf. 0034-913363694. e-mail: guillermo.guardia@upm.es
**Abstract**
Agronomical and environmental benefits are associated with replacing winter fallow by
cover crops (CC). Yet, the effect of this practice on nitrous oxide ($N_2O$) emissions
remains poorly understood. In this context, a field experiment was carried out under
Mediterranean conditions to evaluate the effect of replacing the traditional winter fallow
(F) by vetch (*Vicia sativa* L.; V) or barley (*Hordeum vulgare* L.; B) on greenhouse gas
(GHG) emissions during the intercrop and the maize (*Zea mays* L.) cropping period.
The maize was fertilized following Integrated Soil Fertility management (ISFM)
criteria. Maize nitrogen (N) uptake, soil mineral N concentrations, soil temperature and
moisture, dissolved organic carbon (DOC) and GHG fluxes were measured during the
experiment. The ISFM resulted in low cumulative $N_2O$ emissions (0.57 to 0.75 kg $N_2O$-
N ha$^{-1}$), yield-scaled $N_2O$ emissions (3-6 g $N_2O$-N kg aboveground N uptake$^{-1}$) and N
surplus (31 to 56 kg N ha$^{-1}$) for all treatments. Although CCs increased $N_2O$ emissions
during the intercrop period compared to F (1.6 and 2.6 times in B and V, respectively),
the ISFM resulted in similar cumulative emissions for the CCs and F at the end of the
maize cropping period. The higher C:N ratio of the B residue led to a greater proportion





of $N_2O$ losses from the synthetic fertilizer in these plots, when compared to V. No
significant differences were observed in $CH_4$ and $CO_2$ fluxes at the end of the
experiment. This study shows that the use of both legume and non-legume CCs
combined with ISFM could provide, in addition to the advantages reported in previous
studies, an opportunity to maximize agronomic efficiency (lowering synthetic N
requirements for the subsequent cash crop) without increasing cumulative or yield-
scaled $N_2O$ losses.
**1. Introduction**
Improved resource-use efficiencies are pivotal components of a sustainable
agriculture that meets human needs and protects natural resources (Spiertz, 2010).
Several strategies have been proposed to improve the efficiency of intensive irrigated
systems, where nitrate ($NO_3^-$) leaching losses are of major concern, both during cash
crop and winter fallow periods (Quemada et al., 2013). In this sense, replacing winter
intercrop fallow with cover crops (CCs) has been reported to decrease $NO_3^-$ leaching via
retention of post-harvest surplus inorganic nitrogen (N) (Wagner-Riddle and Thurtell,
1998), consequently improving N use efficiency (NUE) of the cropping system (Gabriel
and Quemada, 2011). Furthermore, the use of CCs as green manure for the subsequent
cash crop may further increase soil fertility and NUE (Tonitto et al., 2006; Veenstra et
al., 2007) through slow release of N and other nutrients from the crop residues, leading
to synthetic fertilizer saving.
From an environmental point of view, N fertilization is closely related with the
production and emission of nitrous oxide ($N_2O$) (Davidson and Kanter, 2014), a
greenhouse gas (GHG) with a molecular global warming potential c. 300 times that of
carbon dioxide ($CO_2$) (IPCC, 2007). Nitrous oxide released from agricultural soils is



mainly generated by nitrification and denitrification processes, which are influenced by
several soil variables (Firestone and Davidson, 1989). Thereby, modifying these
parameters through agricultural management practices (e.g. fertilization, crop rotation,
tillage or irrigation) aiming to optimize N inputs, can lead to strategies for reducing the
emission of this gas (Ussiri and Lal, 2012). In order to identify the most effective GHG
mitigation strategies, side-effects on methane ($CH_4$) uptake and $CO_2$ emission (i.e.
respiration) from soils, which are also influenced by agricultural practices (Snyder et al.,
2009), need to be considered.

To date, the available information linking GHG emission and maize-winter CCs

rotation in the scientific literature is scarce. The most important knowledge gaps include
effects of plant species selection and CCs residue management (i.e. retention,
incorporation or removal) (Basche et al., 2014). Cover crop species may affect $N_2O$
emissions in contrasting ways, by influencing abiotic and biotic soil factors. These
factors include mineral N availability in soil and the availability of carbon (C) sources
for the denitrifier bacterial communities, soil pH, soil structure and microbial
community composition (Abalos et al., 2014). For example, non-legume CCs such as
winter cereals could contribute to a reduction of $N_2O$ emissions due to their deep roots,
which allow them to extract soil N more efficiently than legumes (Kallenbach et al.,
2010). Conversely, the higher C:N ratio of their residues as compared to those of
legumes may provide energy for denitrifiers, thereby leading to higher $N_2O$ losses in the
presence of mineral N from fertilizers (Sarkodie-Addo et al., 2003). Moreover, winter
CCs can also abate indirect gaseous N losses through the reduction of leaching and
subsequent emissions from water resources (Feyereisen et al., 2006). Thus, the
estimated $N_2O$ mitigation potential for winter CCs ranges from 0.2 to 1.1 kg $N_2O$ $ha^{-1}$
$yr^{-1}$ according to Ussiri and Lal (2012).



In a CC-maize rotation system, mineral fertilizer application to the cash crop
could have an important effect on NUE and N losses from the agro-ecosystem. Different
methods for calculating the N application rate (e.g. conventional or integrated) can be
employed by farmers, affecting the amount of synthetic N applied to soil and the overall
effect of CCs on $N_2O$ fluxes. Integrated Soil Fertility Management (ISFM) (Kimani et
al., 2003) provides an opportunity to optimize the use of available resources, thereby
reducing pollution and costs from over-use of N fertilizers (conventional management).
ISFM involves the use of inorganic fertilizers and organic inputs, such as green manure,
aiming to maximize agronomic efficiency (Vanlauwe et al., 2011). When applying this
technique to a CC-maize crop rotation, N fertilization rate for maize is calculated taking
into account the background soil mineral N and the expected available N from
mineralization of CC residues, which depends on residue composition. Differences in
soil mineral N during the cash crop phase may be significantly reduced if ISFM
practices are employed, affecting the GHG balance of the CC-cash crop cropping
system.
Only one study has investigated the effect of CCs on $N_2O$ emissions in
Mediterranean cropping systems (Sanz-Cobena et al., 2014). These authors found an
effect of CCs species on $N_2O$ emissions during the intercrop period. After 4 years of CC
(vetch, barley or rape)-maize rotation, vetch was the only CC species that significantly
enhanced $N_2O$ losses compared to fallow, mainly due to its capacity to fix atmospheric
$N_2$ and because of higher N surplus from the previous cropping phases in these plots. In
this study a conventional fertilization (same N synthetic rate for all treatments) was
applied during the maize phase; how ISFM practices may affect these findings remains
unknown. Moreover, the relative contribution of mineral N fertilizer, CC residues
and/or soil mineral N to $N_2O$ losses during the cash crop has not been assessed yet. In




this sense, stable isotope analysis (i.e. $^{15}$N) has emerged as a way to identify the source
and the dominant processes involved in $N_2O$ production (Arah, 1997). A comprehensive
understanding of the $N_2O$ biochemical production pathways and nutrient sources is
crucial for the development of effective mitigation strategies.
The objective of this study was to evaluate the effect of two different CC species
(barley and vetch) and fallow on GHG emissions during the CC period and during the
following maize cash crop period in an ISFM system. An additional objective was to
study the contribution of the synthetic fertilizer and other N sources to $N_2O$ emissions
using $^{15}$N labelled fertilizer. We hypothesized that: 1) the presence of CCs instead of
fallow would affect $N_2O$ losses, leading to higher emissions in the case of the legume
CC (vetch) in accordance with the studies of Basche et al. (2014) and Sanz-Cobena et
al. (2014); and 2) in spite of the ISFM during the maize period, which theoretically
would lead to similar soil N availability for all plots, the distinct composition of the CC
residues would affect $N_2O$ emissions. In order to test these hypotheses, a field
experiment was carried out using the same management system for 8 years, measuring
GHGs during the 8$^{th}$ year. To gain a better understanding of the effect of the
management practices tested on the overall GHG budget of a cropping system, $CH_4$,
$CO_2$ and yield-scaled $N_2O$ emissions were also analyzed during the experimental
period. The relative contribution of each N source (synthetic fertilizer or soil
endogenous N, including N mineralized from the CCs) to $N_2O$ emissions was also
evaluated by $^{15}$N-labelled ammonium nitrate (AN) in a parallel experiment.

**2. Materials and methods**
*2.1. Site characteristics*





The study was conducted at "La Chimenea" field station (40°03′N, 03°31′W,
550 m a.s.l.), located in the central Tajo river basin near Aranjuez (Madrid, Spain),
where an experiment involving cover cropping systems and conservation tillage has
been carried out since 2006. Soil at the field site is a silty clay loam (*Typic Calcixerept*;
Soil Survey Staff, 2014). Some of the physico-chemical properties of the top 0–10 cm
soil layer, as measured by conventional methods, were: $pH_{H2O}$, 8.16; total organic C,
19.0 g $kg^{-1}$; $CaCO_3$, 198 g $kg^{-1}$; clay, 25%; silt, 49% and sand, 26%. Bulk density of
the topsoil layer determined in intact core samples (Grossman and Reinsch, 2002) was
1.46 g $cm^{-3}$. Average ammonium ($NH_4^+$) content at the beginning of the experiment
was 0.42±0.2 mg N kg $soil^{-1}$ (without differences between treatments). Nitrate
concentrations were 1.5±0.2 mg N kg $soil^{-1}$ in fallow and barley and 0.9±0.1 mg N kg
$soil^{-1}$ in vetch. Initial dissolved organic C (DOC) contents were 56.0±7 mg C kg $soil^{-1}$ in
vetch and fallow and 68.8±5 mg C kg $soil^{-1}$ in barley. The area has a Mediterranean
semiarid climate, with a mean annual air temperature of 14 °C. The coldest month is
January with a mean temperature of 6 °C, and the hottest month is August with a mean
temperature of 24 °C. During the last 30 years, the mean annual precipitation has been
approximately 350 mm (17 mm from July to August and 131 mm from September to
November).
Hourly rainfall and air temperature data were obtained from a meteorological
station located at the field site (CR10X, Campbell Scientific Ltd, Shepshed, UK). A
temperature probe inserted 10 cm into the soil was used to measure soil temperature.
Mean hourly temperature data were stored on a data logger.

*2.2 Experimental design and agronomic management*



Twelve plots (12m × 12m) were randomly distributed in four replications of
three cover cropping treatments, including a cereal and a legume: 1) barley (B)
(*Hordeum vulgare* L., cv. Vanessa), 2) vetch (V) (*Vicia sativa* L., cv. Vereda), and 3)
traditional winter fallow (F). Cover crop seeds were broadcast by hand over the stubble
of the previous crop and covered with a shallow cultivator (5 cm depth) on October $10^{th}$
2013, at a rate of 180 and 150 kg ha$^{-1}$ for B and V, respectively. The cover cropping
phase finished on March $14^{th}$ 2014, with an application of glyphosate (N-
phosphonomethyl glycine) at a rate of 0.7 kg a.e. ha$^{-1}$. All the CC residues were left on
top of the soil. Thereafter, a new set of N fertilizer treatments was set up for the maize
cash crop phase. Maize (*Zea mays* L., Pioneer P1574, FAO Class 700) was direct drilled
on April $7^{th}$ 2014 in all plots, resulting in a plant population density of 7.5 plants m$^{-2}$;
harvesting took place on September $25^{th}$ 2014. The fertilizer treatments consisted of AN
applied on $2^{nd}$ June at three rates: 170, 140 and 190 kg N ha$^{-1}$ in F, V and B plots,
respectively, according to ISFM practices. For the calculation of each N rate, the N
available in the soil (which was calculated following soil analysis as described below),
the expected N uptake by maize crop, and the estimated N mineralized from V and B
residues were taken into account, assuming that crop requirements were 236.3 kg N ha$^{-1}$
(Quemada et al., 2014). Estimated NUE of maize plants for calculating N application
rate was 70% according to the NUE obtained during the previous years in the same
experimental area. Each plot received P as triple superphosphate (45% $P_2O_5$,
Fertiberia®, Madrid, Spain) at a rate of 69 kg $P_2O_5$ ha$^{-1}$, and K as potassium chloride
(60% $K_2O$, Fertiberia®, Madrid, Spain), at a rate of 120 kg $K_2O$ ha$^{-1}$ just before sowing
maize. All N, P and K fertilizers were broadcast by hand, and immediately after N
fertilization the field was irrigated to prevent ammonia volatilization. The main crop




previous to sowing CCs was sunflower (*Helianthus annuus* L., var. Sambro). Neither
the sunflower nor the CCs were fertilized.
In order to determine the amount of $N_2O$ derived from the N fertilizers, double-
labelled AN ($^{15}NH_4^{15}NO_3$, 5 % atom $^{15}N$, from Cambridge Isotope Laboratories, Inc.,
Massachusetts, USA) was applied on 2m x 2m subplots established within each plot at a
rate of 130 kg N ha$^{-1}$. In order to reduce biases due to the use of different N rates (e.g.
apparent priming effects or different mixing ratios between the added and resident soil
N pools) the same amount of N was applied for all treatments. In each subplot, the CC
residue was also left on top of the soil. This application took place on 26$^{th}$ May by
spreading the fertilizer homogenously with a hand sprayer, followed by an irrigation
event.
Sprinkler irrigation was applied to the maize crop in a total amount of 688.5 mm
in 31 irrigation events. Sprinklers were installed in a 12m x 12m framework. The water
doses to be applied were estimated from the crop evapotranspiration (ETc) of the
previous week (net water requirements). This was calculated daily as ETc. = Kc × ETo,
where ETo is reference evapotranspiration calculated by the FAO Penman–Monteith
method (Allen et al., 1998) using data from the meteorological station located in the
experimental field. The crop coefficient (Kc) was obtained using the relationship for
maize in semiarid conditions (Martínez-Cob, 2008).
Two different periods were considered for data reporting and analysis: Period I
(from CC sowing to N fertilization of the maize crop), and Period II (from N
fertilization of maize to the end of the experimental period, after maize harvest).

2.3 *GHG emissions sampling and analyzing*




Fluxes of $N_2O$, $CH_4$ and $CO_2$ were measured from October 2013 to October
2014 using opaque manual circular static chambers as described in detail by Abalos et
al. (2013). One chamber (diameter 35.6 cm, height 19.3 cm) was located in each
experimental plot. The chambers were hermetically closed (for 1 h) by fitting them into
stainless steel rings, which were inserted at the beginning of the study into the soil to a
depth of 5 cm to minimize the lateral diffusion of gases and to avoid the soil disturbance
associated with the insertion of the chambers in the soil. The rings were only removed
during management events. Each chamber had a rubber sealing tape to guarantee an
airtight seal between the chamber and the ring. A rubber stopper with a 3-way stopcock
was placed in the wall of each chamber to take gas samples. Greenhouse gas
measurements were always made with barley/vetch plants inside the chamber. During
the maize period, gas chambers were set up between maize rows.

During Period I, GHGs were sampled weekly or every two weeks. During the

first month after maize fertilization, gas samples were taken twice per week.
Afterwards, gas sampling was performed weekly or fortnightly, until the end of the
cropping period. To minimize any effects of diurnal variation in emissions, samples
were always taken at the same time of the day (10–12 am), that is reported as a
representative time (Reeves et al., 2015).

Measurements of $N_2O$, $CO_2$ and $CH_4$ emissions were made at 0, 30 and 60 min

to test the linearity of gas accumulation in each chamber. Gas samples (100 mL) were
removed from the headspace of each chamber by syringe and transferred to 20 mL gas
vials sealed with a gas-tight neoprene septum. The vials were previously flushed in the
field using 80 mL of the gas sample. Samples were analyzed by gas chromatography
using a HP-6890 gas chromatograph equipped with a headspace autoanalyzer (HT3),
both from Agilent Technologies (Barcelona, Spain). HP Plot-Q capillary columns





transported gas samples to a $^{63}$Ni electron-capture detector (Micro-ECD) to analyze
$N_2O$ concentrations and to a flame ionization detector (FID) connected to a methanizer
to measure $CH_4$ and $CO_2$ (previously reduced to $CH_4$). The temperatures of the injector,
oven and detector were 50, 50 and 350ºC, respectively. The accuracy of the gas
chromatographic data was 1% or better. Two gas standards comprising a mixture of
gases (high standard with 1500 ± 7.50 ppm $CO_2$, 10 ± 0.25 ppm $CH_4$ and 2 ± 0.05 ppm
$N_2O$ and low standard with 200 ± 1.00 ppm $CO_2$, 2 ± 0.10 ppm $CH_4$ and 200 ± 6.00 ppb
$N_2O$) were provided by Carburos Metálicos S.A. and Air Products SA/NV, respectively,
and used to determine a standard curve for each gas. The response of the GC was linear
within 200–1500 ppm for $CO_2$ and 2–10 ppm $CH_4$ and quadratic within 200–2000 ppb
for $N_2O$.
The increases in $N_2O$, $CH_4$ and $CO_2$ concentrations within the chamber
headspace were generally (80% of cases) linear ($R^2$> 0.90) during the sampling period
(1h). Therefore, emission rates of fluxes were estimated as the slope of the linear
regression between concentration and time (after corrections for temperature) and from
the ratio between chamber volume and soil surface area (MacKenzie et al., 1998).
Cumulative $N_2O$, $CH_4$ and $CO_2$, emissions per plot during the sampling period were
estimated by linear interpolations between sampling dates, multiplying the mean flux of
two successive determinations by the length of the period between sampling and adding
that amount to the previous cumulative total (Sanz-Cobena et al., 2014). The
measurement of $CO_2$ emissions from soil including plants in opaque chambers only
includes ecosystem respiration but not photosynthesis (Meijide et al., 2010).

*2.4 $^{15}N$ Isotope analysis*



Gas samples from the subplots receiving double-labelled AN fertilizer were
taken after 60 min static chamber closure 1, 4, 9, 11, 15, 18, 22 and 25 days after
fertilizer application. Stable $^{15}$N isotope analysis of $N_2O$ contained in the gas samples
was carried out on a trace gas analyzer (using cryo-trapping and cryo-focusing) coupled
to a 20/22 isotope ratio mass spectrometer (both from SerCon Ltd., Crewe, UK), at
Rothamsted Research North Wyke. Solutions of 6.6 and 2.9 atom% ammonium
sulphate [(NH$_4$)$_2$SO$_4$] were prepared and used to generate 6.6 and 2.9 atom% $N_2O$
(Laughlin et al., 1997) which were used as reference and quality control standards.
During the experiment, the mean natural abundance of atmospheric $N_2O$ (0.369 atom%
$^{15}$N) was subtracted from measured enriched samples to calculate the atom percent
excess. To obtain the $N_2O$ flux that was derived from fertilizer ($N_2O - N_{dff}$), the Eq. (1)
was used (Loick et al., 2016):
$$N_2O - N_{dff} = N_2O - N \times \left( \frac{N_2O - atom\ percent\ excess_{sample}}{atom\ percent\ excess_{fertilizer}} \right) \quad (1)$$
in which '$N_2O-N$' is the $N_2O$ emission from soil, '$N_2O - ape_{sample}$' is the $^{15}$N
atom% excess of emitted $N_2O$ (being equal to '$^{15}$N atom% of measured samples' minus
0.369 atom% where 0.369 atom% is the mean natural $^{15}$N abundance of '*background*
*$N_2O$*' obtained in our experiment), and '$ape_{fertilizer}$' is the $^{15}$N atom% excess of the
applied fertilizer (Loick et al., 2016).

*2.5 Soil and crop analyses*
In order to relate gas emissions to soil properties, soil samples were collected at
0-10 cm depth during the growing season on almost all gas-sampling occasions,
particularly after each fertilization event. Three soil cores (2.5 cm diameter and 15 cm



length) were randomly sampled close to the ring in each plot, and then mixed and
homogenized in the laboratory. Soil $NH_4^+$ and $NO_3^-$ concentrations were analyzed using
8 g of soil extracted with 50 mL of KCl (1 M), and measured by automated colorimetric
determination using a flow injection analyzer (FIAS 400 Perkin Elmer) provided with a
UV-V spectrophotometer detector. Soil (DOC) was determined by extracting 8 g of
homogeneously mixed soil with 50 mL of deionized water, and analyzed with a total
organic C analyser (multi N/C 3100 Analityk Jena) equipped with an IR detector. The
Water-Filled Pore Space (WFPS) was calculated by dividing the volumetric water
content by total soil porosity. Total soil porosity was calculated according to the
relationship: soil porosity = (1- soil bulk density/2.65), assuming a particle density of
2.65 g cm$^{-3}$ (Danielson and Sutherland, 1986). Gravimetric water content was
determined by oven-drying soil samples at 105 °C with a MA30 Sartorius ®.
Four 0.5m × 0.5m squares were randomly harvested from each plot, before
killing the CC by applying glyphosate. Aerial biomass was cut by hand at soil level,
dried, weighed and ground. A subsample was taken for determination of total N content.
From these samples was determined CC biomass and N contribution to the subsequent
maize.
At maize harvest, two 8 m central rows in each plot were collected and weighed
in the field following separation of grain and straw. For aboveground N uptake
calculations, N content was determined in subsamples of grain and biomass. Total N
content on maize and CC subsamples were determined with an elemental analyzer
(TruMac CN Leco).

*2.6 Calculations and statistical analysis*



Yield-scaled $N_2O$ emissions and N surplus in the maize cash crop were
calculated as the amount of $N_2O$ emitted (considering the emissions of the whole
experiment, i.e. Period I + Period II) per unit of above-ground N uptake, and taking the
difference between N application and above-ground N uptake, respectively (van
Groenigen et al., 2010).
Statistical analyses were carried out with Statgraphics Plus 5.1. Analyses of
variance were performed for all variables over the experiment (except climatic ones),
for both periods indicated in section 2.2. Data distribution normality and variance
uniformity were previously assessed by Shapiro-Wilk test and Levene's statistic,
respectively, and transformed (log10, root-square, arcsin or inverse) before analysis
when necessary. Means of soil parameters were separated by Tukey's honest
significance test at $P<0.05$, while cumulative GHG emissions, YSNE and N surplus
were compared by the orthogonal contrasts method at $P<0.05$. For non-normally
distributed data, the Kruskal–Wallis test was used on non-transformed data to evaluate
differences at $P<0.05$. Linear correlations were carried out to determine relationships
between gas fluxes and WFPS, soil temperature, DOC, $NH_4^+$ and $NO_3^-$. Theses analyses
were performed using the mean/cumulative data of the replicates of the CC treatments
(n=12), and also for all the dates when soil and GHG were sampled, for Period I (n=16),
Period II (n=11) and the whole experimental period (n=27).

**3. Results**
*3.1 Cover crop (Period I)*
*3.1.1 Environmental conditions and WFPS*



Mean soil temperature during the intercrop period was 8.8°C, ranging from 1.8
(December) to 15.5°C (April) (Fig. 1a), which were typical values in the experimental
area. Mean soil temperature during maize cropping period was 24.6°C, which was also
a standard value for this region. The accumulated rainfall during this period was 215
mm, whereas the 30-year mean is 253 mm. Water-Filled Pore Space ranged from 40 to
81% (Fig. 1b). No significant differences were observed for WFPS mean values
between the different treatments ($P>0.05$).

*3.1.2 Mineral N and DOC and cover crop residues*
Topsoil $NH_4^+$ content was below 5 mg N kg soil$^{-1}$ almost of the time in Period I,
although a peak was observed after maize sowing (55 days after CCs kill date) (Fig. 2a),
with the highest values reached in B (50 mg N kg soil$^{-1}$). Mean $NH_4^+$ content was
significantly higher in B than in F ($P<0.05$). Nitrate content increased after CCs killing,
reaching values above 25 mg N kg soil$^{-1}$ in V treatment (Fig. 2c). Mean $NO_3^-$ content
during Period I was significantly higher in the V plots than in the B and F plots
($P<0.001$). Dissolved Organic C ranged from 60 to 130 mg C kg soil $^{-1}$ (Fig. 2e).
Average topsoil DOC content was significantly higher in B than in V and F ($P<0.05$).
The total amount of cover crop biomass left on the ground was 540.5±26.5 and
1106.7±93.6 kg DM ha$^{-1}$ in B and V, respectively. Accordingly, the total N content of
these residues was 11.0±0.6 and 41.3±4.5 kg N ha$^{-1}$ in B and V, respectively.

*3.1.3 GHG fluxes*



Nitrous oxide fluxes ranged from -0.06 to 0.22 mg N m$^{-2}$ d$^{-1}$ (Fig. 3a) in Period
I. The soil acted as a sink for $N_2O$ at some sampling dates, especially for the F plots.
Cumulative fluxes at the end of Period I were significantly greater in CC treatments
compared to F (1.6 and 2.6 higher in B and V, respectively) ($P<0.05$; Table 1). Net $CH_4$
uptake was observed in all intercrop treatments, and daily fluxes ranged from -0.60 to
0.25 mg C m$^{-2}$ d$^{-1}$ (data not shown). No significant differences were observed between
treatments in cumulative $CH_4$ fluxes at the end of Period I ($P>0.05$; Table 1). Carbon
dioxide fluxes (data not shown) remained below 1 g C m$^{-2}$ d$^{-1}$ during the intercrop
period. Greatest fluxes were observed in B although differences in cumulative fluxes
were not significant ($P>0.05$; Table 1). Nitrous oxide emissions were significantly
correlated to $CO_2$ fluxes ($P<0.01$, n=17, r=0.69) and soil temperature ($P<0.05$, n=17,
r=0.55).

*3.2 Maize crop (Period II)*
*3.2.1 Environmental conditions and WFPS*
Mean soil temperature ranged from 19.6 (reached in September) to 32.3°C
(reached in August) with a mean value of 27.9°C (Fig. 1a). Total rainfall during the
maize crop period was 57 mm. Water-Filled Pore Space ranged from 19 to 84% (Fig.
1c). Higher mean WFPS values ($P<0.01$) were measured in B during some sampling
dates.

*3.2.2 Mineral N and DOC*



Topsoil $NH_4^+$ content increased rapidly after N fertilization (Fig. 2b) decreasing

to values below 10 mg N kg soil$^{-1}$ from 15 days after fertilization to the end of the
experimental period. Nitrate concentrations (Fig. 2d) also peaked after AN addition,
reaching the highest value (170 mg N kg soil$^{-1}$) 15 days after fertilization in B ($P<0.05$).
No significant differences ($P>0.05$) between treatments were observed in average soil
$NH_4^+$ or $NO_3^-$ during maize phase. Dissolved Organic C ranged from 56 to 138 mg C kg
soil$^{-1}$ (Fig. 2f). Average topsoil DOC content was 26 and 44% higher in B than in V and
F, respectively ($P<0.001$).

*3.2.3 GHG fluxes, Yield-Scaled $N_2O$ emissions and N surplus*

Nitrous oxide fluxes ranged from 0.0 to 5.6 mg N m$^{-2}$ d$^{-1}$ (Fig. 3b). The highest

$N_2O$ emission peak was observed 1-4 days after fertilization for all plots. Other peaks
were subsequently observed until 25 days after fertilization, particularly in B plots
where $N_2O$ emissions 23 and 25 days after fertilization were higher ($P<0.05$) than those
of F and V (Fig. 3b). No significant differences in cumulative $N_2O$ fluxes were
observed between treatments throughout or at the end of the maize crop period (Table
1), albeit fluxes were numerically higher in B than in V ($0.05<P<0.10$). Daily $N_2O$
emissions were significantly correlated with $NH_4^+$ topsoil content ($P<0.05$, n=12,
r=0.84).

As in the previous period, all treatments were $CH_4$ sinks, without significant

differences between treatments ($P>0.05$; Table 1). Respiration rates ranged from 0.15 to
3.0 g C m$^{-2}$ d$^{-1}$; no significant differences ($P>0.05$; Table 1) were observed among the
$CO_2$ values for the different treatments. Yield-scaled $N_2O$ emissions and N surplus are



shown in Table 1. No significant differences were observed between treatments
although these values were generally lower in V than in B ($0.05<P<0.15$).
Considering the whole cropping period (Period I and Period II), $N_2O$ fluxes
significantly correlated with WFPS ($P<0.05$, n=12, r=0.61), $NH_4^+$ ($P<0.05$, n=27,
r=0.84) and $NO_3^-$ ($P<0.05$, n=27, r=0.50).

*3.2.4 Fertilizer-derived $N_2O$ emissions*
The proportion (%) of $N_2O$ losses from AN, calculated by isotopic analyses, is
represented in Fig. 4. The highest percentages of $N_2O$ fluxes derived from the synthetic
fertilizer were observed one day after fertilization, ranging from 34% (V) to 67% (B).
On average, almost 50% of $N_2O$ emissions in the first sampling event after N synthetic
fertilization came from other sources (i.e. soil endogenous N, including N mineralized
from the CCs). The mean percentage of $N_2O$ losses from synthetic fertilizer throughout
all sampling dates was 2.5 times higher in B compared to V ($P<0.05$). There were no
significant differences between V and F ($P>0.05$).

**4. Discussion**
*4.1 Role of CCs in $N_2O$ emissions: Period I*
Cover crop treatments (V and B) increased $N_2O$ losses compared to F, especially
in the case of V (Table 1). These results are consistent with the meta-analysis of Basche
et al. (2014), which showed that overall CCs increase $N_2O$ fluxes (compared to bare
fallow), with highly significant increments in the case of legumes and a lower effect in



the case of non-legume CCs. In the same experimental area, Sanz-Cobena et al. (2014)
found that V was the only CC significantly affecting $N_2O$ emissions. The greatest
differences between treatments were observed at the beginning (13-40 days after CCs
sowing), and at the end of this period (229 days after CCs sowing) (Fig. 3a). On these
dates, the mild soil temperatures and the relatively high moisture content were more
suitable for soil biochemical processes, which may trigger $N_2O$ emissions (Fig. 1a, b)
(Firestone and Davidson, 1989). Average topsoil $NO_3^-$ was significantly higher in V
(Fig. 2b), which was the treatment that led to the highest $N_2O$ emissions. Legumes such
as V are capable of biologically fixing atmospheric $N_2$, thereby increasing soil $NO_3^-$
content with potential to be denitrified. Further, the mineralization of the most
recalcitrant fraction of the previous V residue (which supplies nearly four times more N
than the B residue, as indicated in section 3.1.2) together with high C-content sunflower
residue could also explain higher $NO_3^-$ contents in V plots (Frimpong et al., 2011), and
higher $N_2O$ losses from denitrification (Baggs et al., 2000). After CCs kill date, N
release from decomposition of roots and nodules and faster mineralization of V residue
compared to that of B (shown by $NO_3^-$ in soil in Fig. 2c) are the most plausible
explanation for the $N_2O$ increases at the end of the intercrop period (Fig. 3a) (Rochette
and Janzen, 2005; Wichern et al., 2008).

Some studies (e.g. Justes et al., 1999; Nemecek et al., 2008) have pointed out

that $N_2O$ losses can be reduced with the use of CCs, due to the extraction of plant-
available N unused by previous cash crop. However, in our study lower $N_2O$ emissions
were measured from F plots without CCs during the intercrop period. This may be a
consequence of higher $NO_3^-$ leaching in F plots (Gabriel et al., 2012; Quemada et al.,
2013), limiting the availability of the substrate for denitrification. Frequent rainfall
during the intercrop period (Fig. 1a) and the absence of N uptake by CCs may have led



to N losses through leaching, resulting in low concentrations of soil mineral N in F
plots.
Nitrous oxide emissions were low during this period, but in the range of those
reported by Sanz-Cobena et al. (2014) in the same experimental area. Total emissions
during Period I represented 8, 10 and 21% of total cumulative emissions in F, B and V,
respectively (Table 1). The absence of N fertilizer application to the soil combined with
the low soil temperatures during winter – which were far from the optimum values for
nitrification and denitrification (25-30 °C) processes (Ussiri and Lal, 2012) – may have
caused these low $N_2O$ fluxes. The significant positive correlation between soil
temperature and $N_2O$ fluxes during this period highlights the key role of this parameter
as a driver of soil emissions (Schindlbacher et al., 2004; García-Marco et al., 2014).

*4.2 Role of CCs in $N_2O$ emissions: Period II*
Isotopic analysis during Period II, in which ISFM was carried out, showed that a
significant proportion of $N_2O$ emissions came from endogenous soil N or the
mineralization of crop residues, especially after the first days following N fertilization
(Fig. 4). In this sense, even though an interaction between crop residue and N fertilizer
application has been previously described (e.g. in Abalos et al., 2013), the similar
proportion of $N_2O$ losses coming from fertilizer in B and F (without residue) one day
after N fertilization revealed the importance of mineral N harbored in soil micropores in
the $N_2O$ bursts after the first irrigation events.
As we hypothesized, although ISFM practices were adopted, the different CCs
played a key role in the $N_2O$ emissions during Period II. Barley plots had higher $N_2O$
emissions than fallow or V-residue plots (at the 10% significance level; Table 1).



Further, a higher proportion of $N_2O$ emissions was derived from the fertilizer in B-
residue than in V-residue plots (Fig. 4). These results are in agreement with those of
Baggs et al. (2003), who reported a higher percentage of $N_2O$ derived from the [15]N-
labeled fertilizer using a cereal (ryegrass) as surface mulching instead of a legume
(bean). The differences between B and V in terms of cumulative $N_2O$ emissions and in
the relative contribution of each source to these emissions (fertilizer- or soil-N) could be
explained by: i) the higher C:N residue of B (20.7±0.7 while that of V was 11.1±0.1,
according to Alonso-Ayuso et al. (2014))  may have provided an energy source for
denitrification (Sarkodie-Addo et al., 2003), increasing the reduction of the $NO_3^-$
supplied by the synthetic fertilizer and enhancing $N_2O$ emissions; ii) $NO_3^-$
concentrations, which tended to be higher in B during the maize cropping phase, could
have led to incomplete denitrification and larger $N_2O/N_2$ ratios (Yamulki and Jarvis,
2002);  iii) the easily mineralizable V residue (with low C:N ratio) provided an
additional N source for soil microorganisms, thus decreasing the relative amount of $N_2O$
derived from the synthetic fertilizer (Baggs et al., 2000; Shan and Yan, 2013); and iv) V
plots were fertilized with a lower amount of immediately available N (i.e. AN) than B
plots, which could have resulted in better synchronization between N release and crop
needs (Ussiri and Lal, 2012) in V plots. Supporting these findings, Bayer et al. (2015)
recently concluded that partially supplying the maize N requirements with winter
legume cover-crops can be considered a $N_2O$ mitigation strategy in subtropical agro-
ecosystems.

The mineralization of B residues resulted in higher DOC contents for these plots

compared to the F or V plots ($P<0.001$). This was observed in both Period I (as a
consequence of soil C changes after the 8-year cover-cropping management) and Period
II (due to the CC decomposition). Although in the present study the correlation between



DOC and $N_2O$ emissions was not significant, positive correlations have been previously
found in other low-C Mediterranean soils (e.g. Vallejo et al., 2006; López-Fernández et
al., 2007). Some authors have suggested that residues with a high C:N ratio can induce
microbial N immobilization (Frimpong and Baggs, 2010, Dendooven et al., 2012). In
our experiment, a $N_2O$ peak was observed in B plots 20-25 days after fertilization (Fig.
3b) after a remarkable increase of $NO_3^-$ content (Fig. 2d), which may be a result of a re-
mineralization of previously immobilized N in these plots.
The positive correlation of $N_2O$ fluxes and soil $NO_3^-$ content and WFPS during
the whole cycle further supports the importance of denitrification process for explaining
$N_2O$ losses in this agro-ecosystem (Davidson et al., 1991; García-Marco et al., 2014).
However, the strong positive correlation of $N_2O$ with $NH_4^+$ indicated that nitrification
was also a major process leading to $N_2O$ fluxes, and showed that the continuous drying-
wetting cycles during a summer irrigated maize crop in a semi-arid region can lead to
favorable WFPS conditions for both nitrification and denitrification processes (Fig. 1c)
(Bateman and Baggs, 2005). Emission Factors ranged from 0.2 to 0.6% of the synthetic
N applied, which were lower than the IPCC default value of 1%. As explained above,
ecological conditions during the intercrop period (rainfall and temperature) and maize
phase (temperature) could be considered as normal (based on the the 30-year average)
in Mediterranean areas. Aguilera et al. (2013) obtained a higher emission factor for high
(1.01%) and low (0.66%) water-irrigation conditions in a meta-analysis of
Mediterranean cropping systems.

*4.3 Methane and CO$_2$ emissions*





As is generally found in non-flooded arable soils, all treatments were net $CH_4$
sinks (Snyder et al., 2009). No significant differences were observed between treatments
in any of the two periods (Table 1), which is similar to the pattern observed by Sanz-
Cobena et al. (2014). Some authors (Dunfield and Knowles, 1995; Tate, 2015) have
suggested an inhibitory effect of soil $NH_4^+$ on $CH_4$ uptake. Low $NH_4^+$ contents during
almost all of the CCs and maize cycle may explain the apparent lack of this inhibitory
effect (Banger et al., 2012). However, during the dates when the highest $NH_4^+$ contents
were reached in V and B (225 days after CCs sowing) (Fig. 3a), $CH_4$ emissions were
significantly higher for these plots (0.12 and 0.16 mg $CH_4$-C m$^{-2}$ d$^{-1}$ for V and B,
respectively) than for F (-0.01 mg $CH_4$-C m$^{-2}$ d$^{-1}$) (data not shown). Similarly, the $NH_4^+$
peak observed two days after fertilization (Fig. 3b) decreased in the order V>F>B, the
same trend as $CH_4$ emissions (which were 0.03, -0.04 and -0.63 mg $CH_4$-C m$^{-2}$ d$^{-1}$ in V,
F and B, respectively; data not shown). Contrary to Sanz-Cobena et al. (2014), the
presence of CCs did not increase $CO_2$ fluxes (Table 1) during Period I (which was
longer than that considered by these authors), even though higher fluxes tended to be
associated to B plots, probably as a consequence of higher root biomass and plant
respiration rates in the cereal (B) than in the legume (V). The decomposition of CC
residues and the growth of maize rooting system resulted in an increase of $CO_2$ fluxes
during Period II (Oorts et al., 2007; Chirinda et al., 2010), although differences between
treatments were not observed.

*4.4 Yield-scaled emissions, N surplus and general assessment*
Yield–scaled $N_2O$ emissions ranged from 1.74 to 7.15 g $N_2O$-N kg aboveground
N uptake$^{-1}$, which is about 1-4 times lower than those reported in the meta-analysis of -



van Groenigen et al. (2010) for a fertilizer N application rate of 150-200 kg ha$^{-1}$. Mean
N surpluses of V and F (Table 1) were in the recommended range (0-50 kg N ha$^{-1}$) by
van Groenigen et al. (2010), while the mean N surplus in B (55 kg N ha$^{-1}$) was also
close to optimal. In spite of higher $N_2O$ emissions in V during Period I (which
accounted for a low proportion of total cumulative $N_2O$ losses during the experiment),
these plots did not emit greater amounts of $N_2O$ per kg of N taken up by the maize
plants, and even tended to decrease YSNE and N surplus (Table 1).
Adjusting fertilizer N rate to soil endogenous N led to lower $N_2O$ fluxes than
previous experiments where conventional N rates were applied (Sanz-Cobena et al.,
2012; Adviento-Borbe et al., 2007), in agreement with the study of Migliorati et al.
(2014). Our results highlight the critical importance of the cash crop period on total $N_2O$
emissions, and demonstrate that the use of either non-legume and –particularly- legume
CCs combined with ISFM may provide an optimum balance between GHG emissions
from crop production and agronomic efficiency (i.e. lowering synthetic N requirements
for a subsequent cash crop, and leading to similar YSNE as a fallow).
The use of CCs has environmental implications beyond effects on direct soil
$N_2O$ emissions. For instance, CCs can mitigate indirect $N_2O$ losses (from $NO_3^-$
leaching). In the study of Gabriel et al. (2012), conducted in the same experimental area,
$NO_3^-$ leaching was reduced (on average) by 30% and 59% in V and B, respectively.
Considering an emission factor of 0.075 from N leached (De Klein et al., 2006), indirect
$N_2O$ losses from leaching could be mitigated by 0.23±0.16  and 0.45±0.17  kg N ha$^{-1}$ yr$^{-}$
$^1$ if V and B are used as CCs, respectively. Furthermore, the recent meta-analysis of
Poeplau and Don (2015) revealed a C sequestration potential of 0.32±0.08 Mg C ha$^{-1}$ yr$^{-}$
$^1$  with the introduction of CCs. These environmental factors together with $CO_2$
emissions associated to CCs sowing and killing, should be assessed in future studies in





order to confirm the potential of CCs for increasing both the agronomic and
environmental efficiency of irrigated cropping areas.

**Conclusions**
Our study confirmed that the presence of CCs (particularly V) during the
intercrop period increased $N_2O$ losses, but the contribution of this phase to cumulative
$N_2O$ emissions considering the whole cropping cycle (intercrop-cash crop) was low (8-
21%). The high influence of the maize crop period over total $N_2O$ losses was not only
due to N synthetic fertilization, but also to CC residue mineralization and especially
endogenous soil N. The type of CC residue determined the N synthetic rate in a ISFM
system and affected the percentage of $N_2O$ losses coming from N fertilizer/soil N as
well as the pattern of $N_2O$ losses during the maize phase (through changes in soil $NH_4^+$,
$NO_3^-$ and DOC concentrations). By employing ISFM, similar $N_2O$ emissions were
measured from CCs and F treatments at the end of the whole cropping period, resulting
in low YSNE (3-6 g $N_2O$-N kg aboveground N uptake$^{-1}$) and N surplus (31 to 56 kg N
ha$^{-1}$). Replacing winter F by CCs did not affect significantly $CH_4$ uptake or respiration
rates neither during intercrop or maize cropping periods. Our results highlight the
critical importance of the cash crop period on total $N_2O$ emissions, and demonstrate that
the use of either legume or non-legume CC combined with ISFM may provide an
optimum balance between GHG emissions from crop production and agronomic
efficiency.

**Acknowledgements**





The authors are grateful to the Spanish Ministry of Economy and Innovation and the
Community of Madrid for their economic support through Projects AGL2012-37815-
C05-01-AGR and the Agrisost-CM Project (S2013/ABI- 2717). We also thank the
technicians and researchers at the Department of Chemistry and Agricultural Analysis
of the Agronomy Faculty (Technical University of Madrid, UPM). Rothamsted
Research is grant funded by the Biotechnology and Biological Sciences Research
Council (BBSRC), UK.

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



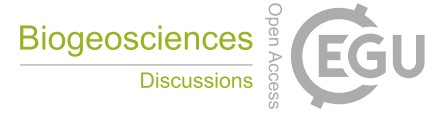


**Table 1** Total cumulative $N_2O$-N, $CH_4$-C and $CO_2$-C fluxes, yield-scaled $N_2O$ emissions (YSNE) and N surplus in the three cover crop treatments (fallow, F, vetch, V, and barley, B) at the end of both cropping periods. $P$ value was calculated with Student's $t$-test and d.f.=9. (*) and S.E. denote significant at $P<0.05$ and the standard error of the mean, respectively.


| Treatment | | $N_2O$ kg $N_2O$-N ha$^{-1}$ | $CH_4$ kg $CH_4$-C ha$^{-1}$ | $CO_2$ kg $CO_2$-C ha$^{-1}$ | Surplus kg N ha$^{-1}$ | YSNE g $N_2O$-N kg aboveground N uptake$^{-1}$ |
|---|---|---|---|---|---|---|
| F | | 0.05 | -0.30 | 443.02 | | |
| V | | 0.13 | -0.28 | 463.01 | | |
| B | | 0.08 | -0.24 | 582.13 | | |
| S.E. | | 0.03 | 0.07 | 46.33 | | |
| **End of Period I** F versus CCs | Estimate | -11.48 | -11.45 | -134.37 | | |
| | t-test | -2.5 | -0.61 | -1.00 | | |
| | P value | 0.03 (*) | 0.56 | 0.34 | | |
| V versus B | Estimate | 5.29 | -6.23 | -127.50 | | |
| | t-test | 1.99 | -0.57 | -1.64 | | |
| | P value | 0.08 | 0.58 | 0.14 | | |
| F | | 0.57 | -0.46 | 2595.07 | 31.47 | 4.21 |
| V | | 0.48 | -0.33 | 2778.84 | 13.72 | 3.06 |
| B | | 0.74 | -0.35 | 2372.07 | 55.94 | 5.64 |
| S.E. | | 0.10 | 0.08 | 177.35 | 15.30 | 0.85 |
| **End of Period II** F versus CCs | Estimate | -7.46 | -23.69 | 83.36 | -3.16 | -0.12 |
| | t-test | -0.30 | -1.25 | 0.19 | -0.08 | -0.14 |
| | P value | 0.77 | 0.24 | 0.86 | 0.94 | 0.89 |
| V versus B | Estimate | -26.59 | 2.08 | 417.8 | -38.67 | -2.59 |
| | t-test | -1.90 | 0.19 | 1.62 | -1.79 | -2.16 |
| | P value | 0.09 | 0.85 | 0.14 | 0.11 | 0.06 |



**Figure captions:**

**Figure 1.** Daily mean soil temperature (°C) rainfall and irrigation (mm) (**a**) and soil WFPS (%) in the three cover crop (CC) treatments (fallow, F, vetch, V, and barley, B) during Period I (**b**) and II (**c**). Vertical lines indicate standard errors.

**Figure 2a, b** $NH_4^+$-N; **c, d** $NO_3^-$-N; and **e, f** DOC concentrations in the 0–10 cm soil layer for the three cover crop (CC) treatments (fallow, F, vetch, V, and barley, B) during both cropping periods. The black arrows indicate the time of spraying glyphosate over the cover crops. The dotted arrows indicate the time of maize sowing. Vertical lines indicate standard errors.

**Figure 3.** $N_2O$ emissions for the three cover crop (CC) treatments (fallow, F, vetch, V, and barley, B) during Period I (**a**) and II (**b**). The black arrows indicate the time of spraying glyphosate over the cover crops. The dotted arrows indicate the time of maize sowing. Vertical lines indicate standard errors.

**Figure 4.** Proportion of $N_2O$ losses (%) coming from N synthetic fertilizer during Period II, for the three cover crop treatments (fallow, F, vetch, V, and barley, B). Vertical lines indicate standard errors. "NS" and * denote not significant and significant at $P<0.05$, respectively.