# Peer review of "Effect of cover crops on greenhouse gas emissions in an irrigated field under"

_Biogeosciences, 2016_

## Author Comment (AC1) · 11 Apr 2016

[Figure]

fig01

[Figure]

fig02

[Figure]

a)

[Figure]

b)

fig03

[Figure]

F vs CCs: NS
V vs B: * (*P*=0.04)

% N$_2$O from N fertilizer

1    4    9    11    15    18    22    25    Average

Days after N fertilization

fig04

---

## Referee Comment (RC1) · Anonymous Referee #2 · 13 Jun 2016

The topic of the manuscript (MS) is within the scope of Biogeosciences and it is generally well written and the results are presented clearly.

However, I have some issues, that need to be addressed by the authors:

1. To me this MS presents rather limited novelty to the study by Sanz-Cobena et al. (2014). Also the added 15N approach brings nothing really new to the current knowledge. The authors should therefore elaborate more clearly the novel and innovative character of their research. 2. What is rather "non-innovative" is the fact, that the cover crops are killed chemically with glyphosate. This is somewhat disappointing for research in agricultural sustainability, as the safe use of glyphosate is under discussion since years. There are alternatives in place also for Mediterranean regions

and might be found among farmers applying organic no-till agriculture. The authors should address this topic in the discussion section, that the application of glyphosate for cover crop management is disputable and alternative measures to remove the cover crops with smart methods are needed (e.g. European project TILMAN-ORG) 3. Cover crop establishment: I am wondering that a hand broadcast technique is used for CC seeding. This might cause too many heterogeneities and influence yield-scaled $N_2O$ emissions. Please discuss. 4. The authors use too many and sometimes unnecessary abbreviations, please adapt. 5. Chambers for GHG sampling: I found it a bit too shallow to insert the stainless rings only 5 cm deep into the soil. There is a high risk of lateral $N_2O$ emission, when the rings/collars are inserted not deep enough (> 10 cm). Please explain.

Recommendation: Major revision

––––––––––––––––––––––––––––

---

## Referee Comment (RC2) · Anonymous Referee #1 · 14 Jun 2016

Guardia et al. report one year of GHG emissions from an irrigated field in semi-arid Spain, testing the effect of intercropping (barley or vetch) versus winter fallow on different cash crops in its 8th year. The soil is calcareous and has a low organic C content. The management of the cash crop (maize) followed the principals of "Integrated Soil Fertility Management", a concept derived from African intercropping practices that takes into account potentially mineralizable N from mulched catch crops when calculating fertilizer requirements for the cash crop. However, unlike I Africa, the catch crop was mulched with glyphosate. The contribution of different N sources to N2O emissions during the cash crop phase was assessed by 15N labelling of fertilizer N.

The concept of "nutrient management" by catch crops (CC) is well explained and

the question how different CC-types affect N2O emissions during and after CC-intercropping is valid and timely. The study is technically well described. Notwithstanding, I have a couple of comments and suggestions, which, I hope, will help to improve the manuscript

L. 1-2: The title "Integrated soil fertility management drives the effect of cover crops on GHG emissions in an irrigated field" is hard to understand, if not misleading; it gives the impression that we are dealing with a "mechanistic" which after all is not the case. Even though the 15N experiment clearly showed that barley residues stimulated N2O emissions from AN fertilizer, the mechanisms behind remain elusive. This is a well conducted descriptive study, which should be reflected in the title. I suggest to change the title.

L. 19: Cumulative N2O emissions were indeed low; but who can say whether this was due to ISFM? It was due to the low fertilization rates, perhaps, but this is not specific for ISFM and there was no control following principals other than ISFM.

L. 19. Cumulative N2O emissions lack time dimension

L. 67-69: This section sounds like making hypotheses after the event; if you want to make a point out of the fact that chemically mulched barley can lead to more N2O emissions during the cash crop phase because it fuels denitrification, offer some explanation why and when you would expect denitrification in a silty clay loam under irrigation. State more precisely that a stimulation of N2O emissions from denitrification by high C/N residues should strictly speaking only occur in the presence of ample nitrate, i.e. right after fertilization.

L. 127 ff.: Soil physico-chemical properties. The soil has a very high pH, high bulk density and low organic carbon. Being in its 8th year of intercropping versus winter fallow, should one expect differences in soil properties among these treatments? And could this explain slight differences in WFPS? Please comment or give soil properties per treatment.

L. 159: Why does ISFM maize with barley as intercrop receive 20 kg more N than with traditional winter fallow? Please explain.

L. 162: How was N mineralization from vetch and barley residues estimated?

L. 170: Would you expect that ammonia volatilization at pH 8.2 differs in plots with and without mulched CCs, even after irrigation? Please comment

L. 220: PLOT columns are primarily for separating inert gases, not for "transporting"

L. 223: replace "detector" with "ECD". The FID is not heated.

L. 243: how was the temperature correction carried out? Opaque chambers deployed for 1 hour in a Mediterranean climate may lead to quite some heating of the chamber air. Did you measure temperatures within the chambers?

L. 256: for equation 1, Senbayram et al. (2009) should be cited and not Loick et al. (2016). Equ. 1 requires the knowledge of 15N atm% excess of emitted N2O (L. 257). This is not equal to the atm% of a sample collected after 1 hour chamber deployment minus the atm% at natural abundance (L. 258)! Senbayram applied this equation to a He-flushed closed flow-through system in which subsampled N2O directly relates to emitted N2O. In the present case, the sample is retrieved from a static chamber in which newly produced N2O mixes with abundant "old" N2O. A Keeling plot approach or some mixing calculation should be applied to derive the true 15N excess of soil emitted N2O before calculating the fraction of N2O derived from AN.

L. 271: Did you filter the extract before DOC analysis? Which pore size?

L. 323: . . . most of the time

L. 325: add that the statistically significant difference in soil ammonium between treatments was found on one sampling date only

L. 330: from figure 2e, it is not obvious that mean DOC contents were higher in B than in V, and if so, the difference was marginal. Besides, ordinary ANOVA on averaged time

series data are not particularly helpful here. Did you use repeated-measure Anova?

L. 344. How can it be that $CO_2$ emissions in plots with intercrops are only insignificantly higher than those in the fallow, of you include plants in your dark chambers and mulch half to 1 tons of dry matter per hectare. Any explanation? Was there a lot of weeds in the fallow? Please give details

L. 388: as outlined above, I believe the absolute numbers for this proportion are wrong. Interestingly, the proportions fluctuated strongly in time but less so across treatments. Did you try to correlate the proportions with any of your ancillary variables (WFPS, temperature, $NO_3$-)?

L. 447: the importance . . . for

L. 447: not clear what you mean by "mineral N harbored in soil micropores"

L. 449: I still don't understand what your finding of larger fertilizer derived $N_2O$ emission in B treatments has to do with ISFM, if ISFM denotes the simple fact that the three treatments received slightly different amounts of fertilizer N. Wouldn't you expect the same without ISFM? L. 491 ff.: include soil pH in the discussion of possible reasons for the overall low emissions and emission factors

L. 536 and 568: optimal balance between GHG emissions and agronomic efficiency provided by ISFM; I do not think you have evidence enough in your data to claim an optimal balance, as long as there is no control experiment receiving equal amounts of mineral fertilizers.

---

## Author Comment (AC2) · 6 Jul 2016

**Response to Reviewer #1 comments**

1. To me this MS presents rather limited novelty to the study by Sanz-Cobena et al. (2014). Also the added 15N approach brings nothing really new to the current knowledge. The authors should therefore elaborate more clearly the novel and innovative character of their research.

We have tried to highlight in the Manuscript the novelty that our study has with respect to Sanz-Cobena et al. (2014). One of the main differences is the use of Integrated Soil Fertility Management (ISFM) in the current study as opposed to conventional fertilization in Sanz-Cobena et al. (2014). The results of the latter study hinted that the effects in soil N availability induced by contrasting cover-crops could represent an opportunity to adjust N fertilization for the cash crop accordingly, without significant yield penalties. This innovative point has now been highlighted in the title ("Effect of cover crops on greenhouse gas emissions in an irrigated field under integrated soil fertility management") and the introduction: "Only one study has investigated the effect of CCs on $N_2O$ emissions in Mediterranean cropping systems (Sanz-Cobena et al., 2014). These authors found an effect of CCs species on $N_2O$ emissions during the intercrop period. After 4 years of CC (vetch, barley or rape)-maize rotation, vetch was the only CC species that significantly enhanced $N_2O$ losses compared to fallow, mainly due to its capacity to fix atmospheric $N_2$ and because of higher N surplus from the previous cropping phases in these plots. In this study a conventional fertilization (same N synthetic rate for all treatments) was applied during the maize phase; how ISFM practices may affect these findings remains unknown."

With regards to the [15]N approach, we agree that there are some previous studies which have evaluated the interactive effects of different crop residues with N synthetic fertilization through [15]N methods (e.g. Baggs et al., 2003; Garcia-Ruiz and Baggs, 2007; Frimpong et al., 2011). Furthermore, [15]N has been used in different cover-cropping experiments (e.g. Bergstrom et al. 2001; Jayasundara et al., 2007; Gabriel and Quemada, 2011, Gabriel et al., 2016) but all of these studies were focused on plant recovery or N leaching. The study of Li et al. (2016) measured [15]$N_2O$ after the application of different CC residues (including roots) and N synthetic fertilizer but under laboratory conditions. To our knowledge, no previous studies have evaluated the relative contribution of CC residues/soil N (which involve the aboveground biomass and the decomposition of root biomass) and N synthetic fertilizers to $N_2O$ emissions under field conditions employing stable isotope techniques. We have elaborated more clearly this novel point in the introduction: "Moreover, the relative contribution of mineral N fertilizer, CC residues and/or soil mineral N to $N_2O$ losses during the cash crop has not been assessed yet. In this sense, stable isotope analysis (i.e. [15]N) represents a way to identify the source and the dominant processes involved in $N_2O$ production (Arah, 1997). Stable Isotope techniques have been used in field studies evaluating N leaching and/or plant recovery in systems with cover crops (Bergström et al., 2001; Gabriel and Quemada, 2011; Gabriel et al., 2016). Furthermore, some laboratory studies have evaluated the effect of different crop residues on $N_2O$ losses using [15]N techniques (Baggs et al., 2003; Li et al., 2016); but to date, no previous studies have evaluated the relative contribution of cover crops (which include the aboveground biomass and the decomposition of root biomass) and N synthetic fertilizers to $N_2O$ emissions under field conditions."

Baggs, E. M., Stevenson, M., Pihlatie, M., Regar, A., Cook, H., and Cadisch, G.: Nitrous oxide emissions following application of residues and fertiliser under zero and conventional tillage. Plant Soil, 254(2), 361-370, 2003.

Bergström, L. F., and Jokela, W. E.: Ryegrass Cover Crop Effects on Nitrate Leaching in Spring Barley Fertilized with (15)NH4(15)NO3. J. Environ. Qual., 30(5), 1659-1667, 2001.

Frimpong, K. A., Yawson, D. O., Baggs, E. M., and Agyarko, K.: Does incorporation of cowpea-maize residue mixes influence nitrous oxide emission and mineral nitrogen release in a tropical luvisol? Nutr. Cycl. Agroecosys., 91(3), 281-292, 2011.

Gabriel, J. L., and Quemada, M.: Replacing bare fallow with cover crops in a maize cropping system: yield, N uptake and fertiliser fate. Eur. J. Agron., 34, 133-143, 2011.

Gabriel, J. L., Alonso-Ayuso, M., García-González, I., Hontoria, C., and Quemada, M.: Nitrogen use efficiency and fertiliser fate in a long-term experiment with winter cover crops. Eur. J. Agron., 79, 14-22, 2016.

Garcia-Ruiz, R., and Baggs, E. M.: $N_2O$ emission from soil following combined application of fertiliser-N and ground weed residues. Plant Soil, 299(1-2), 263-274, 2007.

Jayasundara, S., Wagner-Riddle, C., Parkin, G., von Bertoldi, P., Warland, J., Kay, B., and Voroney, P.: Minimizing nitrogen losses from a corn–soybean–winter wheat rotation with best management practices. Nutr. Cycl. Agroecosys., 79(2), 141-159, 2007.

Li, X., Sørensen, P., Olesen, J. E., and Petersen, S. O.: Evidence for denitrification as main source of $N_2O$ emission from residue-amended soil. Soil Biol. Biochem., 92, 153-160, 2016.

Sanz-Cobena, A., García-Marco, S., Quemada, M., Gabriel, J. L., Almendros, P., and Vallejo, A.: Do cover crops enhance $N_2O$, $CO_2$ or $CH_4$ emissions from soil in Mediterranean arable systems? Sci. Total Environ., 466, 164-174, 2014.

2. What is rather "non-innovative" is the fact, that the cover crops are killed chemically with glyphosate. This is somewhat disappointing for research in agricultural sustainability, as the safe use of glyphosate is under discussion since years. There are alternatives in place also for Mediterranean regions and might be found among farmers applying organic no-till agriculture. The authors should address this topic in the discussion section, that the application of glyphosate for cover crop management is disputable and alternative measures to remove the cover crops with smart methods are needed (e.g. European project TILMAN-ORG).

We agree and are aware that the application of glyphosate is under discussion since years, and now more than ever in the European Union it is a matter under the spotlight. However, the use of non-selective herbicides is a standard and broadly used method followed by conservation tillage growers for cover crop killing in Spain and many other regions. Another alternative for this kind of systems would be mowing but the adequate control is not always achieved, mainly in the case of legumes, in which regrowth is very common. The roller-crimper may be an alternative method but, as well, the legume killing effectiveness is under discussion. Therefore, the glyphosate use seemed an appropriate option that would ensure the killing in both barley and vetch treatments. Moreover, as the study was carried out in a long-term experiment of cover cropping system, it was decided to maintain the same killing method each year. Clearly, further research is needed to investigate this interesting topic, but we considered that it did not fit in any of the subsections of the discussion. Therefore, in the Materials and Methods section of the revised manuscript we have included more information with regards to the use of glyphosate as the killing method in our study: "The cover cropping phase finished on March 14[th] 2014 following local practices, with an application of glyphosate (N-phosphonomethyl glycine) at a rate of 0.7 kg a.e. ha-1. Even though the safe use of glyphosate

is under discussion since years (Chang and Delzell, 2016), it was used in order to preserve the same killing method in all the campaigns in this long-term experiment under conservation tillage management".

Chang, E. T., and Delzell, E.: Systematic review and meta-analysis of glyphosate exposure and risk of lymphohematopoietic cancers. J. Environ. Sci. Heal. B, 51(6), 402-434, 2016.

3. Cover crop establishment: I am wondering that a hand broadcast technique is used for CC seeding. This might cause too many heterogeneities and influence yield-scaled N2O emissions. Please discuss.

In order to reduce economic costs to farmers interested in cover crops, a suitable choice for sowing would be the use of a centrifugal spreader. As the plot size was 12 x 12 $m^2$, the best way to emulate this type of sowing was by hand broadcasting. Results from several previous years and tests showed that this system ensures high homogeneity. Specifically, from cover crop emergence until its killing date, the ground cover was monitored by taking digital photos of four squares (0.5 x 0.5 $m^2$) marked in each plot and lately analyzed with a software based on colorimetry. At the first sampling date (23/10/2013), no differences were observed between vetch samples (ground coverage: 4.3% ± 0.2%), nor in barley (6.7% ± 0.5%).

4. The authors use too many and sometimes unnecessary abbreviations, please adapt.

We thank the reviewer for this remark. Some unnecessary abbreviations, e.g. ammonium nitrate (AN), yield-scaled $N_2O$ emissions (YSNE), N use efficiency (NUE), dry matter (DM) have been removed. If the reviewer thinks that more abbreviations should be removed, we will do it.

5. Chambers for GHG sampling: I found it a bit too shallow to insert the stainless rings only 5 cm deep into the soil. There is a high risk of lateral N2O emission, when the rings/collars are inserted not deep enough (> 10 cm). Please explain.

We thank the reviewer for this comment and we agree that the stainless rings should have been inserted deeper. The rings we used had a height of approximately 10 cm and were inserted into the ground to a depth of ≥5 cm to get a practical height above soil surface of 4-5 cm needed to insert the chamber just above the ground, also preventing water accumulation in the soil surface due to irrigation. We have calculated our average air-filled porosity, which was slightly below 0.3 $cm^3$ $cm^{-3}$. Considering our chamber closure time, the average error may be slightly above 5% (since 6.2 cm is the adequate insertion depth for an air-filled porosity of 0.3 $cm^3$ $cm^{-3}$ and one hour of closure time leading to an error of 5%) (Hutchinson and Livingston, 2001). In further experiments, we will adjust more accurately the insertion depth taking into account our experimental conditions, in order to reduce the error to a minimum.

Hutchinson, G. L., and Livingston, G. P.: Vents and seals in non-steady-state chambers used for measuring gas exchange between soil and the atmosphere. Eur. J. Soil Sci., 52(4), 675-682, 2001.

---

## Author Comment (AC3) · 6 Jul 2016

**Response to Reviewer #2 comments**

L. 1-2: The title "Integrated soil fertility management drives the effect of cover crops on GHG emissions in an irrigated field" is hard to understand, if not misleading; it gives the impression that we are dealing with a "mechanistic" which after all is not the case.

Even though the 15N experiment clearly showed that barley residues stimulated N2O emissions from AN fertilizer, the mechanisms behind remain elusive. This is a well conducted descriptive study, which should be reflected in the title. I suggest to change the title.

We agree with the reviewer's suggestion. We propose a new title more in line with descriptive studies: "Effect of cover crops on greenhouse gas emissions in an irrigated field under integrated soil fertility management".

L. 19: Cumulative N2O emissions were indeed low; but who can say whether this was due to ISFM? It was due to the low fertilization rates, perhaps, but this is not specific for ISFM and there was no control following principals other than ISFM.

We agree with the reviewer that low fertilization rates caused $N_2O$ losses to be low, but these fertilization rates were a consequence of ISFM management, since the crop N requirements were partially supplied through soil inorganic N (measured after the CC killing) and N mineralization, thus reducing the amount of synthetic N. The specific pedo-climatic conditions of our study probably played a role too. The sentence has been changed for better understanding: "Our management (adjusted N synthetic rates due to ISFM) and pedo-climatic conditions resulted…" instead of "The ISFM resulted…"

L. 19. Cumulative N2O emissions lack time dimension

Thanks. This has been corrected (the units are now kg $N_2O$-N ha$^{-1}$ yr$^{-1}$).

L. 67-69: This section sounds like making hypotheses after the event; if you want to make a point out of the fact that chemically mulched barley can lead to more N2O emissions during the cash crop phase because it fuels denitrification, offer some explanation why and when you would expect denitrification in a silty clay loam under irrigation. State more precisely that a stimulation of N2O emissions from denitrification by high C/N residues should strictly speaking only occur in the presence of ample nitrate, i.e. right after fertilization.

We thank the reviewer for this comment and we agree that this point should be better explained. More information and references have been added to this paragraph: "Conversely, it has been suggested that the higher C:N ratio of their residues as compared to those of legumes may provide energy (C) for denitrifiers, thereby leading to higher $N_2O$ losses in the presence of mineral N-$NO_3^-$ from fertilizers (Sarkodie-Addo et al., 2003). In this sense, the presence of cereal residues can increase the abundance of denitrifying microorganisms (Gao et al., 2016), thus enhancing denitrification losses when soil conditions are favorable (e.g. high $NO_3^-$ availability and soil moisture after rainfall or irrigation events, particularly in fine-textured soils) (Stehfest and Bouwman 2006; Baral et al., 2016)".

L. 127 ff.: Soil physico-chemical properties. The soil has a very high pH, high bulk density and low organic carbon. Being in its 8th year of intercropping versus winter fallow, should one expect differences in soil properties among these treatments? And could this explain slight differences in WFPS? Please comment or give soil properties per treatment.

On average, no significant differences between treatments were obtained with regards to soil WFPS. The higher values in B plots in some sampling dates could be a result of increased soil organic matter content in B plots (due to the high C:N residues in this long-term experiment), which could be associated to an enhancement of water-holding capacity (Dabney et al., 2001; Karhu et al., 2011; Hubbard et al., 2013). Since these higher WFPS values were found only in few sampling dates and mean contents did not differ between treatments, we have not discussed these issue in the manuscript, trying to avoid speculative statements.

Soil mineral N and DOC concentrations at the beginning of the experimental period were given in the manuscript for the different treatments. We did not expect differences between treatments in other physico-chemical properties (e.g. pH, texture) due to the different cover cropping treatments in the upper horizon, which was more influenced by the tillage system adopted (conservation tillage). These effects will be evaluated in further campaigns at the same experimental site.

Dabney, S. M., Delgado, J. A., and Reeves, D. W.: Using winter cover crops to improve soil and water quality. Commun. Soil Sci. Plan., 32(7-8), 1221-1250, 2001.

Karhu, K., Mattila, T., Bergström, I., and Regina, K.: Biochar addition to agricultural soil increased CH 4 uptake and water holding capacity–results from a short-term pilot field study. Agric. Ecosyst. Environ., 140(1), 309-313, 2011.

Hubbard, R. K., Strickland, T. C., and Phatak, S.: Effects of cover crop systems on soil physical properties and carbon/nitrogen relationships in the coastal plain of southeastern USA. Soil Till. Res., 126, 276-283, 2013.

L. 159: Why does ISFM maize with barley as intercrop receive 20 kg more N than with traditional winter fallow? Please explain.
L. 162: How was N mineralization from vetch and barley residues estimated?

In order to explain L159 and L162 comments, we will describe in detail the calculation that justifies the choice of different fertilizer doses. For this calculation, the soil inorganic N, N crop requirements, and N mineralization were taken into account as follows:
- Crop requirements ($N_c$) were 236.3 kg N ha$^{-1}$ (Quemada et al., 2014).
- Soil inorganic N ($N_{min}$) was determined to 1-m depth in April, after the CC killing. Values obtained were: fallow = 47.7 kg N ha$^{-1}$; barley = 29.9 kg N ha$^{-1}$; vetch = 45.3 kg N ha$^{-1}$.
For the fallow treatment, the N mineralization ($N_{mineralization}$) considered was 71 kg N ha$^{-1}$, a value observed previous years in the same plots. For barley and vetch treatments, to this value was added the N coming from the mineralization of cover crop residues, estimated as 50% of the cover crop N content. Biomass and %N concentration, necessary to calculate N content, were determined in each cover crop species at the killing moment.
Besides, an efficiency of Nitrogen use efficiency (Ef) of 70% was considered.
Therefore, the rate calculation was as follow:
$$N_f = [N_c - (N_{min} + N_{mineralization})] / Ef$$

$N_f$ fallow = [236.3 – (47.7 +71)]/ 0.7 = 169.3 → 170 kg N ha$^{-1}$
$N_f$ barley = [236.3 – (29.9 +74.6)]/ 0.7 = 188.3 → 190 kg N ha$^{-1}$
$N_f$ vetch = [236.3 – (45.3 +90.5)]/ 0.7 = 143.5 → 140 kg N ha$^{-1}$
Quemada, M., Gabriel, J. L., and Zarco-Tejada, P.: Airborne hyperspectral images and ground-level optical sensors as assessment tools for maize nitrogen fertilization. Remote Sens., 6(4), 2940-2962, 2014.

L. 170: Would you expect that ammonia volatilization at pH 8.2 differs in plots with and without mulched CCs, even after irrigation? Please comment

The presence of mulched CCs could have affected $NH_3$ volatilization, but we think that these losses were small (due to irrigation after fertilization and the type of N source –ammonium nitrate and crop residues-) (Sanz-Cobena et al., 2011; Bittman et al., 2014) with respect to those of $N_2O$, and the differences between treatments were, therefore, negligible.

Bittman, S., Dedina, M., Howard, C.M., Oenema, O., Sutton, M.A., 2014. Options for ammonia mitigation: guidance from the UNECE task force on reactive nitrogen. NERC/Centre for Ecology & Hydrology.

Sanz-Cobena, A., Misselbrook, T., Camp, V., Vallejo, A., 2011. Effect of water addition and the urease inhibitor NBPT on the abatement of ammonia emission from surface applied urea. Atmospheric Environment, 45(8), 1517-1524.

L. 220: PLOT columns are primarily for separating inert gases, not for "transporting"

We thank the reviewer's remark. The sentence has been changed: "Inert gases were separated by HP Plot-Q capillary columns. The gas chromatograph was equipped with a [63]Ni electron-capture detector (Micro-ECD) to analyze $N_2O$ concentrations, and with a flame ionization detector (FID) connected to a methanizer to measure $CH_4$ and $CO_2$ (previously reduced to $CH_4$)".

L. 223: replace "detector" with "ECD". The FID is not heated.

Thanks. The change has been made.

L. 243: how was the temperature correction carried out? Opaque chambers deployed for 1 hour in a Mediterranean climate may lead to quite some heating of the chamber air. Did you measure temperatures within the chambers?

The chambers were all covered with radiant barrier reflective foil. In spite of this covering, the temperature inside the chamber increased compared to the temperature outside the chamber. For this reason, thermometers were placed inside three randomly selected chambers during the closure period of each measurement and the fluxes were corrected for temperature. New information has been included to clarify this point: "The rings were only removed during management events. Each chamber had a rubber sealing tape to guarantee an airtight seal between the chamber and the ring and was covered with a radiant barrier reflective foil to reduce temperature gradients between inside and outside" and "To minimize any effects of diurnal variation in emissions, samples were always taken at the same time of the day (10–12 am), that is reported as a representative time (Reeves et al., 2015). Thermometers were placed inside three randomly selected chambers during the closure period of each measurement and the fluxes were corrected for temperature."

L. 256: for equation 1, Senbayram et al. (2009) should be cited and not Loick et al. (2016).

Ok, we have replaced Loick et al. (2016) by Senbayram et al. (2009).

Equ. 1 requires the knowledge of 15N atm% excess of emitted N2O (L. 257). This is not equal to the atm% of a sample collected after 1 hour chamber deployment minus the atm% at

natural abundance (L. 258)! Senbayram applied this equation to a He-flushed closed flow-through system in which subsampled N2O directly relates to emitted N2O. In the present case, the sample is retrieved from a static chamber in which newly produced N2O mixes with abundant "old" N2O. A Keeling plot approach or some mixing calculation should be applied to derive the true 15N excess of soil emitted N2O before calculating the fraction of N2O derived from AN.

We have followed Senbayram et al. (2009) instructions for the sampling and calculations, and there is no other mixing equation needed. The same equation has been used in several previous studies, such as Lampe et al. 2006 (Sources and rates of nitrous oxide emissions from grazed grassland after application of $^{15}$N-labelled mineral fertilizer and slurry) and Di and Cameron 2008 (Sources of nitrous oxide from $^{15}$N-labelled animal urine and urea fertiliser with and without a nitrification inhibitor, dicyandiamide (DCD)).
The text of Senbayram refers to static chamber as follows: "For ($N_2O$) measurements, PVC chambers (60cm diameter × 25 cm height) were sealed onto the basal rings and gas samples were taken with 12-mL evacuated Exetainers, 0, 20 and 40 minutes after chamber closure."

L. 271: Did you filter the extract before DOC analysis? Which pore size?

Yes, the extract was filtered before DOC analysis using qualitative filter paper 1300/80 (Filter-Lab ®). This information has been added to the manuscript.

L. 323: : : : most of the time

Thanks. This has been corrected.

L. 325: add that the statistically significant difference in soil ammonium between treatments was found on one sampling date only.

Ok, this has been added to the sentence: "Mean $NH_4^+$ content was significantly higher in B than in F ($P<0.05$), but daily $NH_4^+$ concentrations between treatments were only significantly different between treatments in one sampling date (210 days after CCs sowing)".

L. 330: from figure 2e, it is not obvious that mean DOC contents were higher in B than in V, and if so, the difference was marginal. Besides, ordinary ANOVA on averaged time series data are not particularly helpful here. Did you use repeated-measure Anova?

We agree with the reviewer that differences in average contents were small, but with a high level of significance ($P<0.01$, this has been corrected in the text). New information has been included in the paragraph ("Average topsoil DOC content was significantly higher in B than in V and F (10% and 12%, respectively, P<0.01) but differences were only observed in some sampling dates"). We included Fig. 2 as a qualitative and informative representation of the evolution of mineral N and DOC. We tried a repeated-measure ANOVA, but the results did not provide useful information in addition to that of the figure and the average data, besides that the time*treatment interactions complicated the interpretation of the analysis.

L. 344. How can it be that CO2 emissions in plots with intercrops are only insignificantly higher than those in the fallow, of you include plants in your dark chambers and mulch half to 1 tons of dry matter per hectare. Any explanation? Was there a lot of weeds in the fallow? Please give details.

That was an unexpected result. During fall and early winter, low temperatures limited the growth of CCs, and soil respiration rates were small in all treatments. Conversely, from mid-February to the end of CC phase, differences between treatments were higher. We have carried out an ANOVA of average fluxes during this period, and $CO_2$ emissions were significantly higher in B treatment with respect to F, with V showing intermediate values. This information has been added to the text in the Results ("Carbon dioxide fluxes (data not shown) remained below 1 g C $m^{-2}$ $d^{-1}$ during the intercrop period. Greatest fluxes were observed in B although differences in cumulative fluxes were not significant ($P$>0.05; Table 1) in the whole intercrop period, but soil respiration was increased in B, with respect to F, from mid-February to the end of Period I") and Discussion section ("Contrary to Sanz-Cobena et al.(2014), the presence of CCs did not increase $CO_2$ fluxes (Table 1) during the whole Period I (which was longer than that considered by these authors), even though higher fluxes were associated to B (but not V) with respect to F plots in the last phase of the intercrop, probably as a consequence of higher root biomass and plant respiration rates in the cereal (B) than in the legume (V). Differences from fall to early-winter were not significant, since low soil temperatures limited respiration activity").

L. 388: as outlined above, I believe the absolute numbers for this proportion are wrong.
Interestingly, the proportions fluctuated strongly in time but less so across treatments.
Did you try to correlate the proportions with any of your ancillary variables (WFPS, temperature, NO3-)?

Thanks for this remark. Please see our answer to the comment on line 256. Following your suggestion, we have tried to correlate the proportions with these variables. We obtained a significant correlation between DOC content and the proportion coming from the synthetic fertilizer (P<0.05, n=12, r=0.71). These information has been added to the Results ("The mean percentage of $N_2O$ losses from synthetic fertilizer throughout all sampling dates was 2.5 times higher in B compared to V ($P$<0.05) and was positively correlated with DOC concentrations ($P$<0.05, n=12, r=0.71)") and the Discussion section ("the higher C:N residue of B (20.7±0.7 while that of V was 11.1±0.1, according to Alonso-Ayuso et al. (2014)) may have provided an energy source for denitrification (Sarkodie-Addo et al., 2003), favoring the reduction of the $NO_3^-$ supplied by the synthetic fertilizer and enhancing $N_2O$ emissions, as supported by the positive correlation of DOC with the proportion of $N_2O$ coming from the synthetic fertilizer").

L. 447: the importance : : : for

Thanks. This has been corrected.

L. 447: not clear what you mean by "mineral N harbored in soil micropores"

The sentence has been changed. The new sentence is "…revealed the importance of soil mineral N contained in the micropores for the $N_2O$ bursts after the first irrigation events, with respect to the N released from CC residues".

L. 449: I still don't understand what your finding of larger fertilizer derived N2O emission in B treatments has to do with ISFM, if ISFM denotes the simple fact that the three treatments received slightly different amounts of fertilizer N. Wouldn't you expect the same without ISFM?

We agree, the term "ISFM" is unnecessary here. The sentence "As we hypothesized, although ISFM practices were adopted, the different CCs played a key role in the $N_2O$ emissions during

Period II", has been changed to "As we hypothesized, the different CCs played a key role in the $N_2O$ emissions during Period II".

L. 491 ff.: include soil pH in the discussion of possible reasons for the overall low emissions and emission factors

New information about the effect of soil pH on $N_2O$ emissions has been included: "We hypothesized that management practices may have contributed to these low emissions, but other inherent factors such as the high soil pH could have played a role too. Indeed, a higher $N_2O/N_2$ ratio has been associated to acidic soils, so lower $N_2O$ emissions from denitrification could be expected in alkaline soils (Mørkved et al., 2007; Baggs et al., 2010)".

L. 536 and 568: optimal balance between GHG emissions and agronomic efficiency provided by ISFM; I do not think you have evidence enough in your data to claim an optimal balance, as long as there is no control experiment receiving equal amounts of mineral fertilizers.

Thanks for your remark.

The following sentence: "Our results highlight the critical importance of the cash crop period on total $N_2O$ emissions, and demonstrate that the use of either non-legume and –particularly-legume CCs combined with ISFM may provide an optimum balance between GHG emissions from crop production and agronomic efficiency (i.e. lowering synthetic N requirements for a subsequent cash crop, and leading to similar YSNE as a fallow)" has been deleted from de Manuscript.

New information and references about the effect of adjusting N synthetic rate has been added:

"Adjusting fertilizer N rate to soil endogenous N led to lower $N_2O$ fluxes than previous experiments conducted under similar environmental conditions where conventional N rates were applied (e.g. Adviento-Borbe et al., 2007; Hoben et al., 2011; Sanz-Cobena et al., 2012; Li et al., 2015), in agreement with the study of Migliorati et al. (2014). Moreover, $CO_2$ equivalent emissions associated to manufacturing and transport of N synthetic fertilizers (Lal, 2004) can be reduced when low synthetic N input strategies, such as ISMF, are employed"

The second sentence (in the Conclusions) has been changed for better understanding: "Our results highlight the critical importance of the cash crop period on total $N_2O$ emissions, and demonstrate that the use of non-legume and –particularly– legume CCs combined with ISFM could be considered an efficient practice from both environmental and agronomic points of view, leading to similar $N_2O$ losses per kilogram of aboveground N uptake as fallow".

---

## Author Comment (AC4) · 6 Jul 2016

We thank the anonymous referees for their time and their efforts to improve the science and the presentation of our work. In the attached pdf documents, we respond to each of the referees' comments. Please note that because of the structure of the Biogeosciences Discussions review process, we do not provide a revised manuscript at this stage but instead provide an indication of where and how we will revise the paper, if given the chance.